# Inhibition of Autophagy Promotes Hemistepsin A-Induced Apoptosis via Reactive Oxygen Species-Mediated AMPK-Dependent Signaling in Human Prostate Cancer Cells

**DOI:** 10.3390/biom11121806

**Published:** 2021-12-01

**Authors:** Kwang-Youn Kim, Un-Jung Yun, Seung-Hee Yeom, Sang-Chan Kim, Hu-Jang Lee, Soon-Cheol Ahn, Kwang-Il Park, Young-Woo Kim

**Affiliations:** 1Korean Medicine (KM) Application Center, Korea Institute of Oriental Medicine, 70 Cheomdan-ro, Dong-gu, Daegu 41062, Korea; lokyve@kiom.re.kr; 2School of Korean Medicine, Dongguk University, Gyeongju 38066, Korea; yun2456@dongguk.ac.kr (U.-J.Y.); jungdaeman95@naver.com (S.-H.Y.); 3Medical Research Center, College of Oriental Medicine, Daegu Haany University, Gyeongsan 38610, Korea; sckim@dhu.ac.kr; 4College of Veterinary Medicine, Gyeongsang National University, Jinju 52828, Korea; hujang@gnu.ac.kr; 5Department of Microbiology & Immunology, Pusan National University School of Medicine, Yangsan 50612, Korea

**Keywords:** Hemistepsin A, AMPK, ROS, autophagy, apoptosis, prostate cancer

## Abstract

Chemotherapy is an essential strategy for cancer treatment. On the other hand, consistent exposure to chemotherapeutic drugs induces chemo-resistance in cancer cells through a variety of mechanisms. Therefore, it is important to develop a new drug inhibiting chemo-resistance. Although hemistepsin A (HsA) is known to have anti-tumor effects, the molecular mechanisms of HsA-mediated cell death are unclear. Accordingly, this study examined whether HsA could induce apoptosis in aggressive prostate cancer cells, along with its underlying mechanism. Using HsA on two prostate cancer cell lines, PC-3 and LNCaP cells, the cell analysis and in vivo xenograft model were assayed. In this study, HsA induced apoptosis and autophagy in PC-3 cells. HsA-mediated ROS production attenuated HsA-induced apoptosis and autophagy after treatment with N-acetyl-L-cysteine (NAC), a ROS scavenger. Moreover, autophagy inhibition by 3-MA or CQ is involved in accelerating the apoptosis induced by HsA. Furthermore, we showed the anti-tumor effects of HsA in mice, as assessed by the reduced growth of the xenografted tumors. In conclusion, HsA induced apoptosis and ROS generation, which were blocked by protective autophagy signaling.

## 1. Introduction

Prostate cancer is the second most commonly diagnosed cancer in men, accounting for more than one in five new cancer diagnoses [1,2,3]. Recently, it has been reported that several studies showed cancer incidence and mortality rates have increased [4,5]. Although several new therapies for prostate cancer have been attempted, the fundamental mechanisms of prostate cancer need to be clarified [6,7,8].

Various studies have shown that apoptosis and autophagy are important mechanisms for cancer cell death [7]. Apoptosis is considered a major mechanism for cancer cell death, but these mechanisms do not block chemo-resistant cancer and advanced cancer [8]. Recently, many studies have revealed autophagy as a new and important target mechanism for chemo-resistant cancer therapies [9]. Many cancer cells induce autophagy to block apoptosis [8,10,11].

Autophagy can be initiated by activation of the AMP-activated protein kinase (AMPK) pathway, resulting in oxidative stress, such as reactive oxygen species (ROS) [12,13,14,15]. Recently, it has been shown that ROS are crucial factors in communications between apoptosis and autophagy [8]. Several reports have shown that AMPK is a major cellular homeostasis regulator; activating this pathway can enhance autophagy in cancer cells [16]. Previous studies have shown that targeting AMPK to regulate autophagy may be a new theoretical strategy for treating cancer [17].

Several studies have shown that the chemical constituents of *Hemistepta lyrata* (Bunge) Bunge are sesquiterpene lactones, including Hemistepsin A (HsA), hemistepsin B, isoamberboin, and 8-hydroxyzaluzanin [18,19,20]. Many studies have reported the diverse pharmacological activities of HsA, such as its protective effects against liver injury [18,21,22], anticancer effects against colorectal and hepatocarcinoma, and cell cycle arrest [23,24,25].

This study examined the induction of autophagy by HsA in chemo-resistant prostate cancer cells as a protective role of apoptosis. HsA induced cytotoxicity via autophagy and apoptosis induction. In addition, ROS production and AMPK activation are important mediators in HsA-induced autophagy. Furthermore, xenografted tumor models showed that the anti-tumor effect of HsA reduced tumor growth. Hence, inhibition of autophagy may be a new strategy to treat chemo-resistant prostate cancer.

## 2. Materials and Methods

### 2.1. Chemicals and Reagents

The Hemistepsin A (HsA) standard was kindly provided by Prof. Jong Rok Lee (Daegu Haany University, Korea) [20]. 3-Methyladenine (3-MA) and N-acetyl-cysteine (NAC) were purchased from Sigma (Sigma, St. Louis, MO, USA). The AMPK inhibitor compound C was obtained from Calbiochem (Shanghai, China). Antibodies of AMPKα1, acetyl-CoA carboxylase (ACC), β-actin, rabbit/mouse horseradish peroxidase (HRP)-conjugated IgG were acquired from Santa Cruz Biotechnology (Santa Cruz, CA, USA). Anti-LC3B, β-actin, phospho-AMPKα (Thr 172), and phospho-ACC (Ser 79) antibodies were supplied by Cell Signaling Tech (Denver MA). The enhanced chemiluminescence (ECL) western blot reagent kit was procured from Pierce (Rockford, IL, USA).

### 2.2. Cell Lines and Cell Culture

Human prostate cancer cells lines, PC-3 and LNCaP cells were obtained from the American Type Culture Collection (ATCC, Manassas, VA, USA). These cells were maintained at 37 °C in a humidified atmosphere containing 5% CO_2_.

### 2.3. Cell Viability Assay

The cells were seeded in 48-well plates at a density of 2 × 10^5^ cells per well as previously described [10,12,25]. The cells were detected at 540 nm using a VERSAmax microplate reader (Molecular Devices, Sunnyvale, CA, USA). The cell viability was defined relative to the untreated control (viability (% control) = 100 × absorbance of treated sample/absorbance of control).

### 2.4. Annexin V/Propidium Iodide (PI) Assay

A fluorescein isothiocyanate (FITC)-Annexin V Apoptosis Detection Kit (BD Biosciences, San Jose, CA, USA) was used to detect the apoptotic cells according to the manufacturer’s protocol as previously described [10,12,25]. The stained cells were analyzed by flow cytometry (FACSVerse, Becton Dickinson, Franklin Lakes, NJ, USA), and the apoptotic cells were calculated using Cell Quest Pro software on Mac^®^ OS 9 (Becton Dickinson).

### 2.5. Detection of Acidic Vesicular Organelles

To analyze the acidic vesicular organelles (AVOs), human prostate cancer PC-3 and LNCaP cells were cultured in a glass-bottom dish as previously described [10,12,25]. Cells were stained with 1 μM acridine orange at 37 °C, and analyzed by flow cytometry (FACSVerse, Becton Dickinson, Franklin Lakes, NJ, USA).

### 2.6. Immunofluorescence for LC-3

The levels of LC-3 expression were determined using immunofluorescence analysis as previously described [10,12,25]. DAPI was used to stain the nuclei. The samples were visualized using a laser scanning confocal microscope (Olympus FluoView FV1000).

### 2.7. Measurement of Intracellular ROS Generation

The level of intracellular ROS production was determined. PC-3 cells were cultured in 6-well plates at a density of 1 × 10^4^ cells/well. The cells were treated with HsA, incubated with 10 µM 2′-7′dichlorofluorescin diacetate (DCFH-DA) at 37 °C for 30 min, and washed twice with PBS. The cells were analyzed by flow cytometry ((FACSVerse) and quantified using Cell Quest Pro software on Mac^®^ OS 9 (Becton Dickinson).

### 2.8. In Vivo Tumor Growth Analysis

PC-3 cells (2 × 10^6^) were suspended in 100 μL PBS and inoculated subcutaneously in the right flank of six-week-old BALB/C nude mice (Orient, Busan, Korea). After tumor formation, the mice were divided randomly into 3 groups, which consisted of vehicle (PBS) control (*n* = 8) and 5 mg/kg (*n* = 8) and 10 mg/kg (*n* = 8) HsA treatment administered orally daily by Oral Zonde. After three weeks of treatment, the mice were sacrificed. The tumor sizes were measured weekly to observe the dynamic changes in tumor growth, which were calculated using a standard formula: volume = (length × width2)/2. The bodyweight of the mice was measured weekly to evaluate the systemic toxicity of the drug. All animal experimental procedures were approved and monitored by Institutional Animal Care and Use Committee in Dongguk University.

### 2.9. Western Blot

Western blot analysis was performed as previously described [10,12,25]. The extracted protein sample mixed with a 5 × loading buffer (250 mM Tris-HCl pH 6.8, 5% 2-Mercaptoethanol, 10% SDS, 0.5% Bromophenol blue, 50% Glycerol) was boiled for 10 min at 95 °C. Detection was performed using ECL Supersingnal West Pico Chemiluminescent Substrate according to the manufacturer’s instruction.

### 2.10. Statistical Analysis

The experiments were repeated at least three times with consistent results. Unless stated otherwise, the data are expressed as the mean ± SD. ANOVA was used to compare the experimental groups with the control values. Comparisons between multiple groups were performed using a Tukey’s multiple comparison tests. The results were statistically significant at * *p* < 0.05.

## 3. Results

### 3.1. Hemistepsin A Inhibits Cell Growth and Induces Apoptosis in Human Prostate Cancer Cells

The effects of HsA on cell viability in prostate cancer PC-3 and LNCaP cells were first examined using a MTT assay. Treatment with increasing HsA concentrations inhibited the cell viability of both PC-3 and LNCaP cells in a dose- and time-dependent manner (Figure 1A,B). Moreover, HsA exhibited potential anticancer activity. The induction of apoptosis by HsA was analyzed by flow cytometry in the cells stained by Annexin V-FITC and PI. HsA increased the early apoptotic and late apoptotic cells in a dose-dependent manner in PC-3 and LNCaP cells (Figure 1C). Consistent with the MTT assay, PC-3 cells were more apoptotic resistant than LNCaP cells. Furthermore, we treated with HsA of the indicated concentration and then evaluated the levels of apoptosis-related proteins, including Bcl-xL, caspase-3, and PARP using western blot. The results clearly showed that procaspase-3 was suppressed, and PARP showed more cleavage by HsA in a dose-dependent manner (Figure 1D).

### 3.2. Hemistepsin A Induces Autophagy in Human Prostate Cancer Cells

The potential role of autophagy in HsA-induced cytotoxicity in prostate cancer cells was evaluated to determine the underlying mechanisms. First, the accumulation of acidic vesicular organelles (AVOs) was confirmed by HsA stained with acridine orange (AO) by flow cytometry and confocal microscopy. HsA increased the accumulation of AVOs in both PC-3 and LNCaP cells in a dose-dependent manner (Figure 2A). Furthermore, PC-3 cells showed more accumulation of AVOs than LNCaP cells, which is opposite to the apoptosis results. The formation of AVOs in the HsA-treated cells was confirmed by the larger increase in the red fluorescence levels than the vehicle-treated cells (Figure 2B, left panel). Furthermore, immunofluorescence and western blot analysis confirmed the autophagy-specific markers, microtubule-associated protein 1 light chain 3B (LC3B), and Beclin-1. As expected, punctuates of GFP-LC3B were enhanced by HsA in PC-3 cells (Figure 2B, right panel). Furthermore, the levels of LC3B-II were increased and Beclin-1 was decreased by HsA in a dose-dependent manner in both PC-3 and LNCaP cells (Figure 2C). Moreover, the ratio of LC3B-II/LC3B-I was also increased by HsA in the LNCaP cells (i.e., vehicle, 1 fold; HsA 10 μM, 1.13 ± 0.17 fold; HsA 20 μM, 1.09 ± 0.14 fold).

### 3.3. Inhibition of Autophagy Enhances Hemistepsin A-Induced Apoptosis in Human Prostate Cancer PC-3 Cells

The HsA-induced autophagy was further confirmed using an autophagy inhibitor, 3-methyladenine (3-MA) or chloroquine (CQ). The 3-MA treatment decreased the levels of HsA-induced AVOs accumulation, but CQ increased them, indicating that HsA could induce autophagy in PC-3 cells (Figure 3A). Furthermore, 3-MA and CQ enhanced HsA-induced apoptosis significantly in PC-3 cells. Interestingly, treatment with 3-MA alone also induced minor cell apoptosis (Figure 3B). A pretreatment with 3-MA reduced the conversion of LC-3B-I/II, whereas pretreatment with CQ enhanced the conversion (Figure 3C,D). Furthermore, caspase-3 activation and PARP cleavage were also decreased by a pretreatment with 3-MA and CQ (Figure 3C,D). Overall, HsA-induced autophagy has protective roles against apoptosis and facilitates HsA-induced apoptosis by inhibiting autophagy.

### 3.4. Activation of AMPK Mediates Hemistepsin A-Induced Autophagy and Apoptosis in Human Prostate Cancer PC-3 Cells

The underlying mechanism of AMP-activated protein kinase (AMPK)-mediated autophagy induction in HsA-treated prostate cancer cells was examined. HsA increased AMPKα1, ACC, and LKB1 phosphorylation significantly in PC-3 cells in a time-dependent manner, which demonstrated potently induced AMPK activation (Figure 4A). Furthermore, compound C, an AMPK inhibitor, largely suppressed HsA-induced autophagy by reducing the conversion of LC3B-I/II in PC-3 cells (Figure 4B), while enhancing HsA-induced apoptosis (Figure 4C). These results suggest that the activation of AMPK by HsA-mediates autophagy and apoptosis-resistance in prostate PC-3 cells.

### 3.5. Hemistepsin A-Induced Autophagy and Apoptosis Is Associated with ROS-Mediated AMPK Activation

Several studies have shown that ROS are important regulators in chemotherapeutic agent-mediated apoptosis in many cell types. This study examined whether ROS production is related to HsA-induced apoptosis of PC-3 cells. The level of ROS was measured at various doses of the HsA treatment by measuring the fluorescence intensity of DCH-DA-stained cells. The cellular ROS level was increased after treatment of the HsA for 24 h (Figure 5A,B). As expected, pretreatment with a ROS scavenger, N-acetyl cysteine (NAC), decreased the level of ROS induced by HsA (Figure 5B). Furthermore, the NAC stimulations reduced the autophagic and apoptotic cells significantly (Figure 5C,D) and pretreatment with NAC restored HsA-induced caspase-3 activation, LC3B, and PARP cleavage (Figure 5E). HsA-induced AMPK activation was recovered by the NAC pretreatment (Figure 5F). Overall, the production of cellular ROS might be related to AMPK activation as well as autophagy and apoptosis induction of PC-3 cells.

### 3.6. Hemistepsin A Inhibits Tumor Growth In Vivo

Xenograft studies were conducted using PC-3 cells in male BALB/c nude mice. The tumor growth was delayed in the HsA treatment group (5 mg/kg and 10 mg/kg) compared to the vehicle group (Figure 6A,B). The change in body weight was similar in the HsA treatment group and vehicle group (Figure 6C). These results demonstrated an anti-tumor effect of HsA against prostate cancer in vivo.

## 4. Discussion

Herbal medicine is one of the alternative strategies for cancer therapy. It is difficult to find active compounds in the herbs because some fluent components in the herbs are sometimes not active in the effects of beneficial action. Here, we examined the anticancer effect of HsA on human prostate cancer cells using in vitro and in vivo approaches. The data suggested that the underlying molecular mechanisms involve autophagy-dependent apoptosis via ROS-mediated AMPK signaling.

Hemistepsin A is a sesquiterpene lactone isolated from the aerial parts and flowers of *Hemistepta lyrata* (Bunge) Bunge [19]. Some studies have shown that HsA has potent anticancer effects by molecular mechanisms, such as downregulation of STAT3 [23], inhibition of pyruvated dehydrogenase kinase activity [24], G0/G1-phase cell cycle arrest, and AMPK and p53/p21 signaling activation in many cancer cell lines [25]. Furthermore, previous reports have shown that HsA inhibits hepatitis and liver fibrosis [21]. HsA also manages liver fibrosis by inhibiting NF-κB/Akt-dependent signaling [18].

Interestingly, HsA has different effects on the cell type. The apoptotic rates of PC-3 cells were lower than in LNCaP cells (Figure 1). On the other hand, the autophagic rates showed opposite results to the apoptosis results (Figure 2). Overall, HsA induced-autophagy has cytoprotective roles involved in apoptosis resistance.

Various studies have shown that autophagy is a potent mechanism for eliminating incorrect folded or aggregated proteins and maintaining the energy homeostasis to respond to stress [26,27]. Although the roles of crosstalk between apoptosis and autophagy have been studied [28,29], autophagy has been a widely used therapy for chemo-resistant cancer [30,31,32,33]. In this study, HsA induced autophagy by converting LC-3B. As expected, 3-MA, an autophagy inhibitor, inhibited HsA-induced autophagy. The inhibition of autophagy increased HsA-induced apoptosis, which suggest that autophagy affects apoptosis-resistant cancer cells.

A line of reports has shown that crosstalk between apoptosis and autophagy is associated with the production of ROS [34,35]. Recent studies have shown that representative molecules, such as PI3K/AKT/mTOR and AMPK signaling pathways regulate autophagy and apoptosis induction [36,37]. Moreover, the AMPK pathway is essential for autophagy initiation under cellular stress [38,39]. This study indicated that HsA modulates ROS production in PC-3 cells. On the other hands, NAC, a ROS scavenger, attenuates HsA-mediated ROS production and prevents the activation of AMPK and ACC, resulting in autophagy inhibition.

Intracellular ROS production has vital roles in AMPK activation under cellular circumstances such as apoptosis and autophagy [40,41,42]. In this study, autophagy and LC3 expression were restored by the ROS inhibition, and the inhibition of ROS production led to a decrease in HsA-induced apoptosis (Figure 5). Moreover, the inactivation of AMPK by compound C enhanced HsA-induced apoptosis (Figure 4). The inactivation of AMPK decreased the HsA-induced autophagy by compound C (Figure 4). The inhibition of ROS generation linked to AMPK and ACC was restored in PC-3 cells. These results suggest that HsA-induced autophagy triggers the ROS-mediated AMPK pathways.

The in vitro results were reconfirmed in vivo in the experimental xenograft mouse model with PC-3 cells on apoptosis and autophagy induced by HsA in prostate cancer cells. HsA potently inhibited tumor growth without any bodyweight changes, suggesting that the HsA treatment did not have toxic side effects (Figure 6).

## 5. Conclusions

HsA might be a potential anticancer agent combined with autophagy inhibitors for patients with prostate cancer because it can regulate apoptosis and autophagy via ROS-mediated AMPK activation. These findings suggest the therapeutic use of HsA as a potential strategy for prostate cancer treatment.

## Figures and Tables

**Figure 1 biomolecules-11-01806-f001:**
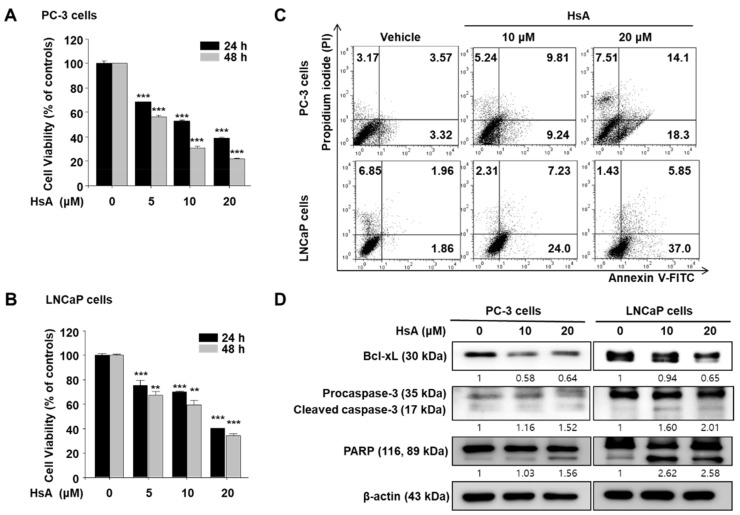
Hemistepsin A (HsA) inhibits cell growth and induces apoptosis in prostate cancer cells. (**A**,**B**) Cell growth inhibition: Cell viability was assessed by 3-(4,5-dimethyl-thiazol-2-yl)-2,5-diphenyl-etrazolium bromide (MTT) assay. (**C**) Apoptosis induction: Cells were treated with various concentrations of HsA for 24 h. (**D**) Bcl-xL and Pro-caspase-3 and poly (ADP-ribose) polymerase (PARP) expression. All data are representative of at least three independent experiments. Data show mean ± SD. ** *p* < 0.01. *** *p* < 0.001 compared with the control group.

**Figure 2 biomolecules-11-01806-f002:**
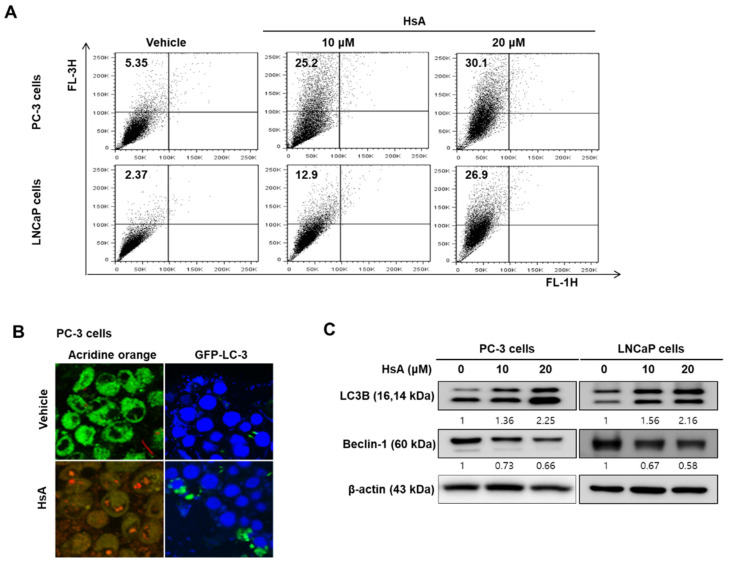
Hemistepsin A (HsA) induces autophagy in prostate cancer cells. (**A**,**B**) Acidic vacuoles’ detection: After treatment with HsA for 24 h, cells were stained with 1 μM acridine orange (AO) for 20 min. Then, cells were analyzed by flow cytometry (**A**) and visualized using a laser scanning confocal microscope by AO staining (**B**). (**C**) LC-3 and Beclin-1 expression: After treatment with HsA, total cell lysates were subjected to SDS-PAGE for western blot analysis.

**Figure 3 biomolecules-11-01806-f003:**
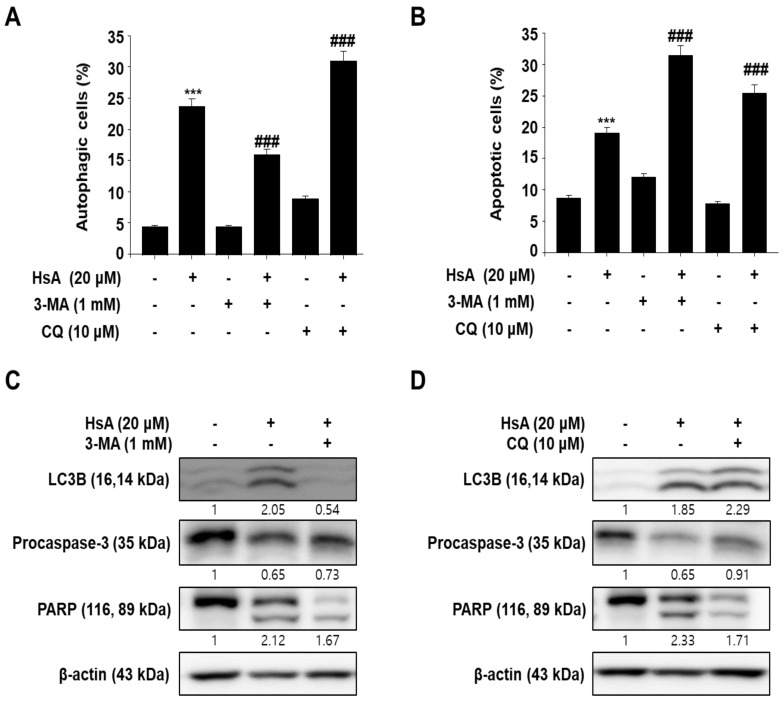
Autophagy inhibition enhances Hemistepsin A (HsA)-induced apoptosis in prostate cancer PC-3 cells. (**A**) Effects of 3-MA and Chloroquine (CQ) in HsA-induced autophagy: Acidic vesicular organelles (AVOs) were detected by flow cytometry. (**B**) Effects of 3MA and CQ on HsA-induced apoptosis: Apoptotic cells were detected by flow cytometry. (**C**) Effects of 3-MA on LC-3, pro-caspase-3 and PARP protein expression. (**D**) Effects of CQ on LC-3, pro-caspase-3 and PARP protein expression. All data are representative of at least three independent experiments. Data show mean ± SD. *** *p* < 0.001 compared with the control group; ### *p* < 0.001 compared with the HsA treated group.

**Figure 4 biomolecules-11-01806-f004:**
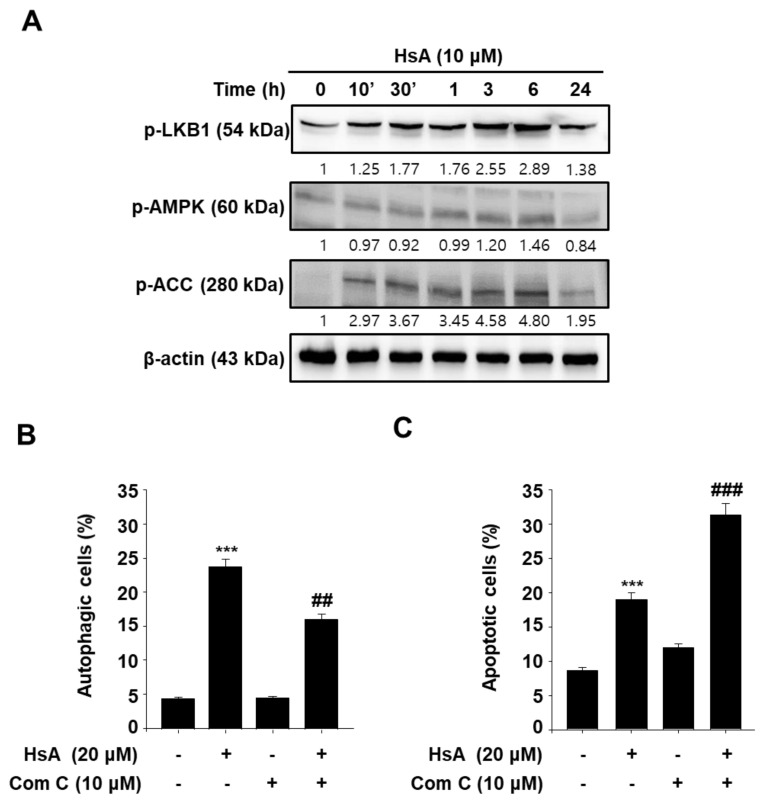
AMPK activation regulates Hemistepsin A (HsA)-induced autophagy and apoptosis in prostate cancer PC-3 cells. (**A**) Western blot analysis for p-AMPK, p-ACC and p-LKB1. PC-3 cells were treated with 20 μM HsA for the indicated times. (**B**) Effect of Compound C on HsA-induced autophagy. (**C**) Effect of Compound C on HsA-induced apoptosis. After the treatment with Compound C for 1 h, PC-3 cells were treated with 20 μM HsA. Autophagic and apoptotic cells were detected by flow cytometry. All data are representative of at least three independent experiments. Data show mean ± SD. *** *p* < 0.001 compared with the control group; ## *p* < 0.05, ### *p* < 0.001 compared with the HsA treated group.

**Figure 5 biomolecules-11-01806-f005:**
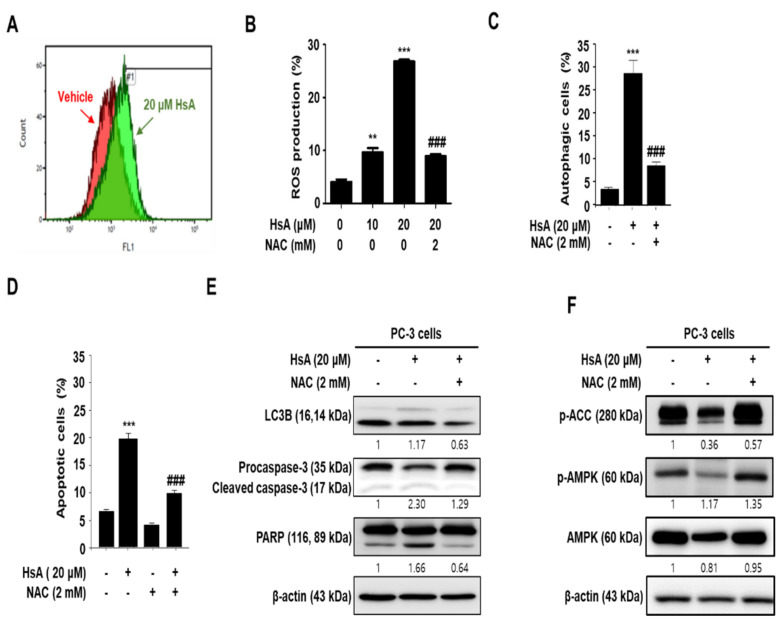
The production of ROS regulates Hemistepsin A (HsA)-induced autophagy and apoptosis in prostate cancer PC-3 cells. (**A**) Effects of HsA on ROS production. (**B**) Effects of NAC on HsA-induced ROS production. (**C**) Effects of NAC on HsA-induced autophagy. (**D**) Effects of NAC in on HsA-induced apoptosis. (**E**) Effects of NAC on LC-3, caspase-3 and PARP protein expression. (**F**) Effects of NAC on p-ACC and p-AMPK activation: Cells were treated with 20 μM HsA for 24 h, with or without 2 mM NAC, and then incubated with 1 μg/mL DCF-DA in serum-free medium for 20 min. Apoptotic and autophagic cells or DCF-DA-positive cells were detected by flow cytometry. Protein expressions were detected by western blot analysis. All data are representative of at least three independent experiments. Data show mean ± SD. ** *p* < 0.05, *** *p* < 0.001 compared with the control group; ### *p* < 0.001 compared with the HsA treated group.

**Figure 6 biomolecules-11-01806-f006:**
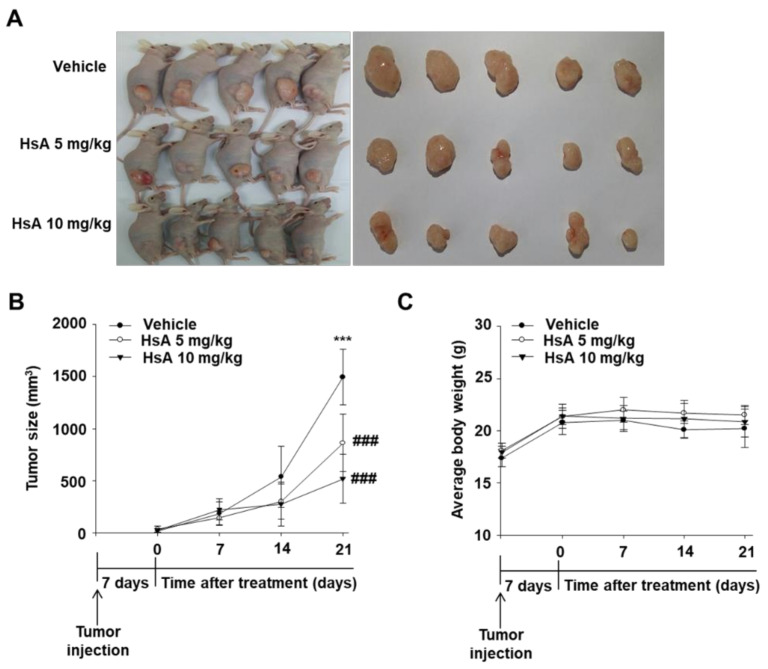
Hemistepsin A (HsA) inhibits the growth of human prostate cancer xenograft in vivo. (**A**,**B**) Tumor volume. (**C**) Mouse body weight. *** *p* < 0.001 compared with the 0 day; ### *p* < 0.001 compared with the HsA treated group.

## Data Availability

The data presented in this study are available on request from the corresponding author.

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
