# Peer review of "Inhibition of Autophagy Promotes Hemistepsin A-Induced Apoptosis via Reactive Oxygen Species-Mediated AMPK-Dependent Signaling in Human Prostate Cancer Cells"

_biomolecules, 2021, doi:10.3390/biom11121806_

Round 1

Reviewer 1 Report

In this publication, Kim et al. dissect the effects of Hemistepsin A on prostate cancer cell apoptosis and autophagy through reactive oxygen species-mediated AMPK-dependent signaling. While the topic is relevant and fairly analyzed there are several issued that should be solved:

Specific points:

Figure 1D: Changes in procaspase 3 and cleaved caspase 3 from this Figure are not evident. Additionally, the signal for procaspase-3 is over-saturated and lower exposure image is suggested. Better images should be provided and quantified when differences are hard to detect by eye like in this case.

If both 3-MA and CQ are inhibitors of autophagy, why do they have opposite effects?

Section 3.5: the authors say: ''HsA increased AMPKα1, ACC, and LKB1 phosphorylation significantly in PC-3 cells in a time-dependent manner, which demonstrated potently induced AMPK activation (Figure 4A).'' However, Figure 5F is showing the opposite. How is this explained?

Section 3.6: would it be possible to provide IHC images of dissected tumors which show apoptosis/autophagy markers?

Minor points:

In Abstract, it is written: ''Chemotherapy is an essential strategy for cancer treatment. On the other hand, consistent exposure to chemotherapeutic drugs induces chemo-resistance in cancer cells through a variety of mechanisms.'' Here, the connection with HsA is missing; it should be either briefly stated how is HsA important in the light of chemotherapy resistance, or these sentences could be omitted.

In Introduction the authors say: ''Several new therapies for prostate cancer have been attempted, including new drugs and combination therapies to target chemo-resistant and advanced metastatic cancers [6].'' Reference [6] is from 2010; along with this reference, a more recent reference could be cited.

In Introduction the authors say: ''This study examined the induction of autophagy by HsA in chemo-resistant prostate cancer cells as a protective role of apoptosis.'' It should be clarified how PC-3 and LNCaP cells are considered to be chemoresistant.

Section 2.7: it should be briefly explained what DCFH-DA is and the abbreviation should be explained.

Section 2.8: ''After tumor formation, the mice were divided randomly into two groups of three…'' It is unclear here how many mice were used per group.

Section 2.9: ''The extracted protein sample was boiled for 5 min in a fivetimes loading buffer.'' What loading buffer was used?

Section 3.1: ''evacuated by Western blot'' should be ''evaluated by…''

Section 3.2: it could be briefly mentioned what AVOs are

Figure 2B and Figure 5A, B, C, D: Neither on Figure nor in Figure captions it isn't stated which cell line was used.

Figure 3C and D and where applicable: procaspase-3 and caspase-3 WBs should be shown separately, or the whole image (without cropping) should be provided

Figure 3C and D and Figure 5 captions: ''Effects of 3-MA in LC-3…'' Instead of ''in'' it should be written ''on''

Discussion: the authors say: ''More recently, autophagy has been a widely used therapy for chemoresistant cancer [28, 29].'' References [28, 29] are not recent.

Conclusions: The authors say: ''These findings suggest the therapeutic use of HsA as a potential strategy for bladder cancer treatment''. Instead of ''bladder'' it should be written ''prostate''.

The information on statistical significance could be added on graphs.

Author Response

Specific points:

1. Figure 1D: Changes in procaspase 3 and cleaved caspase 3 from this Figure are not evident. Additionally, the signal for procaspase-3 is over-saturated and lower exposure image is suggested. Better images should be provided and quantified when differences are hard to detect by eye like in this case.

We thank the reviewer for the thoughtful comments. We agree about that and so changed caspase 3 in the figure 1D

2. If both 3-MA and CQ are inhibitors of autophagy, why do they have opposite effects?

We thank the reviewer for the thoughtful insights.

Chloroquine (CQ) autophagy inhibitor increased the formation of Acidic vesicular organelles (AVOs) [ref. #1]. CQ is frequently used classic inhibitor of autophagy that blocks the binding of autophagosomes to lysosomes by altering the acidic environment of lysosomes, resulting in inducing autophagosomes accumulation. The autophagosome was also identified by with AVO and Autophagosomes accumulation can increase AVOs formation. In reference #1, CQ also increased AVOs though CQ inhibited autophagy. In the same manner, CQ increased AVOs and increased apoptosis by reducing autophagy in this study. We incorporated in “Discussion” section.

Ref. #1. Kazuhito Sasaki, et al., Chloroquine potentiates the anti-cancer effect of 5-fluorouracil on colon cancer cells, BMC Cancer, 10: 370 (2010) 

3. Section 3.5: the authors say: ''HsA increased AMPKα1, ACC, and LKB1 phosphorylation significantly in PC-3 cells in a time-dependent manner, which demonstrated potently induced AMPK activation (Figure 4A).'' However, Figure 5F is showing the opposite. How is this explained?

In the Figure 4A, 'HsA increased AMPKα1, ACC, and LKB1 phosphorylation significantly in PC-3 cells in a time-dependent manner but significantly decreased pAMPK at 24h inducing apoptosis. In the Figure 5F, we incubated with HsA for 24h and investigated by immunoblotting. pAMPK and pACC were decreased by HsA at 24h and NAC inhibited effect of HsA by blocking decrease of pAMPK and pACC.

4. Section 3.6: would it be possible to provide IHC images of dissected tumors which show apoptosis/autophagy markers?

Although we agreed the reviewer’s recommendation, we don't stock tumor tissues from mouse, unfortunately. However, the volume of the tumor was significantly inhibited by HsA.

Minor points:

5. In Abstract, it is written: ''Chemotherapy is an essential strategy for cancer treatment. On the other hand, consistent exposure to chemotherapeutic drugs induces chemo-resistance in cancer cells through a variety of mechanisms.'' Here, the connection with HsA is missing; it should be either briefly stated how is HsA important in the light of chemotherapy resistance, or these sentences could be omitted.

We thank the reviewer for the thoughtful insights and suggestions. According to your suggestion, we incorporated in abstract “Chemotherapy is an essential strategy for cancer treatment. On the other hand, consistent exposure to chemotherapeutic drugs induces chemo-resistance in cancer cells through a variety of mechanisms. Therefore, It is important to develop and apply a new drug that inhibits resistance against anti-cancer drugs”.

6. In Introduction the authors say: ''Several new therapies for prostate cancer have been attempted, including new drugs and combination therapies to target chemo-resistant and advanced metastatic cancers [6].'' Reference [6] is from 2010; along with this reference, a more recent reference could be cited.

As the reviewer mentioned, we cited more recent reference.

7. In Introduction the authors say: ''This study examined the induction of autophagy by HsA in chemo-resistant prostate cancer cells as a protective role of apoptosis.'' It should be clarified how PC-3 and LNCaP cells are considered to be chemoresistant.

A human prostate cancer cell line PC3 is resistant to camptothecin (CPT) and LNCaP cells are more resistant to docetaxel than DU145 and PC3 cells. We cited with reference.

Yukihiro Akao, et. al, High expression of sphingosine kinase 1 and S1P receptors in chemotherapy-resistant prostate cancer PC3 cells and their camptothecin-induced up-regulation, Biochem Biophys Res Commun., 21;342(4):1284-90 (2006)

Lu Gan, et. al, Resistance to docetaxel-induced apoptosis in prostate cancer cells by p38/p53/p21 signaling. The Prostat, 1;71 (11) :1158-1166 (2011)

8. Section 2.7: it should be briefly explained what DCFH-DA is and the abbreviation should be explained.

We explained that, thank you for your suggestion.

9. Section 2.8: ''After tumor formation, the mice were divided randomly into two groups of three…'' It is unclear here how many mice were used per group.

As the reviewer suggested, we incorporated that in section 2.8.

10. Section 2.9: ''The extracted protein sample was boiled for 5 min in a fivetimes loading buffer.'' What loading buffer was used?

As the reviewer suggested, we incorporated that in section 2.9.

11. Section 3.1: ''evacuated by Western blot'' should be ''evaluated by…''

We changed the sentence to “We treated with HsA of indicated concentration and then evaluated the levels of apoptosis-related proteins, including Bcl-xL, caspase-3, and PARP using western blot”.

12. Section 3.2: it could be briefly mentioned what AVOs are

We incorporated that.

13. Figure 2B and Figure 5A, B, C, D: Neither on Figure nor in Figure captions it isn't stated which cell line was used.

We mentioned in Figure legend and figure.

14. Figure 3C and D and where applicable: procaspase-3 and caspase-3 WBs should be shown separately, or the whole image (without cropping) should be provided

We changed that.

15. Figure 3C and D and Figure 5 captions: ''Effects of 3-MA inLC-3…'' Instead of ''in'' it should be written ''on''

We corrected it.

16. Discussion: the authors say: ''More recently, autophagy has been a widely used therapy for chemoresistant cancer [28, 29].'' References [28, 29] are not recent.

Thanks for your suggestion. We changed to more recent references.

17. Conclusions: The authors say: ''These findings suggest the therapeutic use of HsA as a potential strategy for bladder cancer treatment''. Instead of ''bladder'' it should be written ''prostate''.

As a reviewer recommended, we changed it

18. The information on statistical significance could be added on graphs.

As a reviewer recommended, we added statistical significance.

Reviewer 2 Report

Overall, the research work in the manuscript sounds logical, and the results are meaningful. However, I think there are few major/minor points (as below) need to be addressed by the authors before the manuscript could further consider for publication in Biomolecules.

  1. Fig 1-C, how the authors possess the apoptosis flow data? The pics didn’t show the whole cells, most of the cells are in the edge. For the cells around 10^0, I think that are the cell debris. The results should be re-analysis.
  2. Fig 1-D, I have question about the bands of pro-caspase-3 and cleaved caspase-3, pro-caspase-3 is around 35KD and cleaved caspase-3 are 17KD and 19KD, the authors use 10% SDS-polyacrylamide gel, the bands of markers should not so close to each other.
  3. Fig 2-A, the cell debris should be excluded and the result need to re-analysis.
  4. Fig legend of fig 2, for pic A, there didn’t show and microscope pics, ‘visualized using with a laser scanning confocal microscope’ should be moved to legend B.
  5. Fig 3-A, when treated with compound CQ which is an autophagy inhibitor, there are around 10% autophagic cells while control group has 5%. Please explain.
  6. Fig 3-C and D, the change of cleaved caspase 3 are not obvious, please calculate the fold change of them.
  7. Fig 4-B, why choose Hela as a comparison?
  8. What is Compound C? is it a commodity? Or the product from a lab? Is there any references support that compound C is a AMPK inhibitor?
  9. No scale on fig 6-A.
  10. In 2.8, there should be 3 group, not two group. The drugs were administered orally, is the drug were added in their drinking water? how long will the mice take all drugs? How to confirm the mice take all the daily dosage?
  11. How to choose the time point to treat with drugs? In Fig 6-B, why after 7 day of tumor injection, 0 day, the tumor size are nearly 0? The mice are small when inject the tumor, since all the mice are around 18g, usually we choose the mice around 25g.
  12. In the fig 6 legend, it said the mice received daily oral administration for 5 weeks, which against methods 2.8 and fig 6-B and C.
  13. In general, the language is fluent, yet there are still some mistakes emerged within the article. For example: Page 6, line 51 and 53 (also refer to line 81 &82 in the article) “and their” is not complete. Same as Page 6 line 61 (also refer to line 85 in the article) “used to”.

Author Response

Overall, the research work in the manuscript sounds logical, and the results are meaningful. However, I think there are few major/minor points (as below) need to be addressed by the authors before the manuscript could further consider for publication in Biomolecules.

1. Fig 1-C, how the authors possess the apoptosis flow data? The pics didn’t show the whole cells, most of the cells are in the edge. For the cells around 10^0, I think that are the cell debris. The results should be re-analysis.

We thank the reviewer for the thoughtful comments. To exclude debris and cell aggregates, we set appropriate FSC vs SSC gates according to the manufacturer’s instructions. Therefore, the cells around 10^0 are not the cell debris.

2. Fig 1-D, I have question about the bands of pro-caspase-3 and cleaved caspase-3, pro-caspase-3 is around 35KD and cleaved caspase-3 are 17KD and 19KD, the authors use 10% SDS-polyacrylamide gel, the bands of markers should not so close to each other.

It is a similar question to comment 1of reviewer 1. We changed caspase 3 in the figure 1D.

3. Fig 2-A, the cell debris should be excluded and the result need to re-analysis.

We set appropriate FSC vs SSC gates to exclude debris and cell aggregates likewise Figure1-C.

4. Fig legend of fig 2, for pic A, there didn’t show and microscope pics, ‘visualized using with a laser scanning confocal microscope’ should be moved to legend B.

We thank the reviewer for the kind comments. We changed legend of Figure 2B as below.

(A and B) Acidic vacuoles' detection: After treatment with Hemistepsin A for indicated dose for 24 h, cells were stained with 1 μM acridine orange (AO) at 37°C in the dark for 20 min. Then, cells were washed with PBS and, subsequently, analyzed by flow cytometry (A) and visualized using with a laser scanning confocal microscope (B).

5. Fig 3-A, when treated with compound CQ which is an autophagy inhibitor, there are around 10% autophagic cells while control group has 5%. Please explain.

It is a similar question to comment 2 of reviewer #1. Please read the answer to comment 2 of reviewer #1 as below.

  • Chloroquine (CQ) autophagy inhibitor increased the formation of Acidic vesicular organelles (AVOs) [ref. #1]. CQ is frequently used classic inhibitor of autophagythat blocks the binding of autophagosomes to lysosomes by altering the acidic environment of lysosomes, resulting in inducing autophagosomes accumulation. The autophagosomewas also identified by with AVO and Autophagosomes accumulation can increase AVOs formation. In reference #1, CQ also increased AVOs though CQ inhibited autophagy. In the same manner, CQ increased AVOs and increased apoptosis by reducing autophagy in this study. We incorporated in “Discussion” section.

Ref. #1. Kazuhito Sasaki, et al., Chloroquine potentiates the anti-cancer effect of 5-fluorouracil on colon cancer cells, BMC Cancer, 10: 370 (2010) 

6. Fig 3-C and D, the change of cleaved caspase 3 are not obvious, please calculate the fold change of them.

We changed caspase 3 blot and incorporated the calculated fold change in the figure 3C and 3D.

7. Fig 4-B, why choose Hela as a comparison?

Hela cell is LKB1-deficient. Therefore, it was used negative control for AMPK phosphorylation.

8. What is Compound C? is it a commodity? Or the product from a lab? Is there any references support that compound C is a AMPK inhibitor?

Compound C is commonly used as an inhibitor of AMPK. We obtained from Calbiochem as mentioned in the “materials and methods” [Ref. #2 and #3].

Ref #2. Xiaona Liu, Rishi Raj Chhipa, Ichiro Nakano and Biplab Dasgupta, The AMPK Inhibitor Compound C Is a Potent AMPK-Independent Antiglioma Agent, Mol Cancer Ther; 13(3) March 596-605 (2014)

Ref #3. Weng-Lang Yang,William Perillo BS, Deanna Liou HS, Philippe Marambaud ,Ping Wang M, AMPK inhibitor compound C suppresses cell proliferation by induction of apoptosis and autophagy in human colorectal cancer cells, Journal of Surgical Oncology 2012:106: 680-688.

9. No scale on fig 6-A.

Although we agreed the reviewer’s recommendation, we don't stock tumor tissues from mouse, unfortunately. However, the volume of the tumor was significantly inhibited by HsA.

10. In 2.8, there should be 3 group, not two group. The drugs were administered orally, is the drug were added in their drinking water? how long will the mice take all drugs? How to confirm the mice take all the daily dosage?

In 2.8 section, we corrected to “After tumor formation, the mice were divided randomly into three group, which consisted of vehicle (PBS) control and 5 and 10 mg/kg HsA treatment administered orally daily.”

11. How to choose the time point to treat with drugs? In Fig 6-B, why after 7 day of tumor injection, 0 day, the tumor size are nearly 0? The mice are small when inject the tumor, since all the mice are around 18g, usually we choose the mice around 25g.

Four-week-old male BALB-c nu/nu mice were housed in filtered-air flow cabinets for one week. Five-week-old BALB-c nu/nu mice were about 18g. Empirically, the young mouse forms tumors well. Each animal was injected subcutaneously with 2 × 106 A431 cells. Once the tumors Once the tumors became palpable (about 50 mm3), vehicle (PBS) control and HsA administered orally daily. The smaller the tumor is, the better the drug's effect is.

12. In the fig 6 legend, it said the mice received daily oral administration for 5 weeks, which against methods 2.8 and fig 6-B and C.

We are sorry for our mistake, and corrected 5weeks to 3weeks.

13. In general, the language is fluent, yet there are still some mistakes emerged within the article. For example: Page 6, line 51 and 53 (also refer to line 81 &82 in the article) “and their” is not complete. Same as Page 6 line 61 (also refer to line 85 in the article) “used to”.

We thank the reviewer for the thoughtful comments.

Reviewer 3 Report

Inhibition of Autophagy Promotes Hemistepsin A-Induced Apoptosis via Reactive Oxygen Species-Mediated AMPK-Dependent Signaling in Human Prostate Cancer Cells

In this manuscript author elucidate the role of HsA in regulation of autophagy and apoptosis in AMPK dependent manner. I have the following suggestions that must be addressed:

1.     The author should plot the viability plots of PC-3 and LNCap cells together to determine the relative cell vitality of the cells.

2.     Statement “The induction of apoptosis by AnnexinV-FITC and PI double staining was analyzed by flow cytometry” need to be corrected. Apoptosis is not induced by AnnexinV.

3.     Figures 1A and B: There is no stander deviation (provide the statical significance for each group with respective control)

4.     Figure 1D. cleaved caspase-3 activity is not reflected as it is mentioned in the result section. It is preferable to leave out the cleaved caspase 3.

5.     Figure 2A-C: The explanation for the outcome is incorrect. Induction of autophagy results in the conversion of LC3B-I to LC3B-II.LC3B-II is a lipidation state that fuses with autophagosomes to enable autophagy-mediated cargo breakdown in the lysosome. In this case, I propose examining the LC-3BI/LC-3B-II ratio. LC-3B-II does not merge with the lysosome during autophagy suppression and inhibiting autophagosome fusion with the lysosome results in accumulation of both LC3-I and LC-3B-II. It will be useful to examine the expression of p62/SQSTM1 in order to confirm the influence of HsA on autophagy.

  1. Statement “A pretreatment with 3-MA reduced the conversion of LC-3B-I/II, whereas pretreatment with CQ enhanced the conversion (Fig. 3C and D)”. There is a mountain of evidence that chloroquine inhibits autophagy by preventing autophagosome fusion with the lysosome, resulting in the accumulation of both LC-3B I and LC-3B-II. (For further information, see PMID 17611390).

  1. Statement “Overall, HsA-induced autophagy has protective roles against apoptosis and facilitates HsA-induced apoptosis by inhibiting autophagy.” The outcome is contradictory and must be rewritten.

  1. Figure 5 F treatment of HsA alone causes the decrees in the phosphorylation of p-AMPK which is contradicting the previous result.

  1. Please add western blot of total protein (AMPK,)

  1. The manuscript is lacking novelty; the Author should have sufficient know-how to interpret the exact reasons of the research outcome. The author is unable to explain how stimulation of autophagy by HsA facilitate a protective effect while triggering apoptosis.

  1. There is a mountain of evidence indicating autophagy induction promotes tumor cell survival under stress conditions and autophagy suppression reduces tumorigenesis (Ref PMID 33946505). However, autophagy is a contest-dependent process that is affected by the stage of tumorigenesis. I have a recommendation to include this point in the discussion.

Author Response

In this manuscript author elucidate the role of HsA in regulation of autophagy and apoptosis in AMPK dependent manner. I have the following suggestions that must be addressed:

1. The author should plot the viability plots of PC-3 and LNCap cells together to determine the relative cell vitality of the cells.

In the Figure 1C, we compared HsA-induced apoptosis in PC-3 and LNCap cell. PC-3 cells were more sensitive than LNcap cells. We changed the sentences as the recommendation in the Result Section of the new MS.

2. Statement “The induction of apoptosis by AnnexinV-FITC and PI double staining was analyzed by flow cytometry” need to be corrected. Apoptosis is not induced by AnnexinV.

Thank you for your comment. We corrected to “Apoptosis was stained Annexin V and PI and measured with flow cytometry”

3. Figures 1A and B: There is no stander deviation (provide the statical significance for each group with respective control)

We added statistical significane.

4. Figure 1D. cleaved caspase-3 activity is not reflected as it is mentioned in the result section. It is preferable to leave out the cleaved caspase 3.

We changed caspase 3 in the figure 1D

5. Figure 2A-C: The explanation for the outcome is incorrect. Induction of autophagy results in the conversion of LC3B-I to LC3B-II.LC3B-II is a lipidation state that fuses with autophagosomes to enable autophagy-mediated cargo breakdown in the lysosome. In this case, I propose examining the LC-3BI/LC-3B-II ratio. LC-3B-II does not merge with the lysosome during autophagy suppression and inhibiting autophagosome fusion with the lysosome results in accumulation of both LC3-I and LC-3B-II. It will be useful to examine the expression of p62/SQSTM1 in order to confirm the influence of HsA on autophagy.

We agreed the reviewer’s comments. The ratio of LC3B-II/LC3B-I was also increased by HsA in the LNCaP cells [i.e., vehicle, 1 fold; HsA 10 uM, 1.13 fold; HsA 20 uM, 1.09 fold].

6. Statement “A pretreatment with 3-MA reduced the conversion of LC-3B-I/II, whereas pretreatment with CQ enhanced the conversion (Fig. 3C and D)”. There is a mountain of evidence that chloroquine inhibits autophagy by preventing autophagosome fusion with the lysosome, resulting in the accumulation of both LC-3B I and LC-3B-II. (For further information, see PMID 17611390).

We agreed the comments. We already changed the sentences. It is a similar question to comment 2 of reviewer #1. Please read the answer to comment 2 of reviewer #1 as below.

  • Chloroquine (CQ) autophagy inhibitor increased the formation of Acidic vesicular organelles (AVOs) [ref. #1]. CQ is frequently used classic inhibitor of autophagythat blocks the binding of autophagosomes to lysosomes by altering the acidic environment of lysosomes, resulting in inducing autophagosomes accumulation. The autophagosomewas also identified by with AVO and Autophagosomes accumulation can increase AVOs formation. In reference #1, CQ also increased AVOs though CQ inhibited autophagy. In the same manner, CQ increased AVOs and increased apoptosis by reducing autophagy in this study. We incorporated in “Discussion” section.

Ref. #1. Kazuhito Sasaki, et al., Chloroquine potentiates the anti-cancer effect of 5-fluorouracil on colon cancer cells, BMC Cancer, 10: 370 (2010) 

7. Statement “Overall, HsA-induced autophagy has protective roles against apoptosis and facilitates HsA-induced apoptosis by inhibiting autophagy.” The outcome is contradictory and must be rewritten.

 We corrected it to “HsA-induced autophagy has protective roles against apoptosis and so autophagy inhibition can increase sensitivity to HsA by increasing apoptosis.”

8. Figure 5 F treatment of HsA alone causes the decrees in the phosphorylation of p-AMPK which is contradicting the previous result.

 It is a similar question to comment 3 of reviewer 1. Please read it and refer to it as below.

In the Figure 4A, 'HsA increased AMPKα1, ACC, and LKB1 phosphorylation significantly in PC-3 cells in a time-dependent manner but significantly decreased pAMPK at 24h inducing apoptosis. In the Figure 5F, we incubated with HsA for 24h and investigated by immunoblotting. pAMPK and pACC were decreased by HsA at 24h and NAC inhibited effect of HsA by blocking decrease of pAMPK and pACC

9. Please add western blot of total protein (AMPK,) 

We added western blot of total protein (AMPK,) in Figure 5E and 5F. 

10. The manuscript is lacking novelty; the Author should have sufficient know-how to interpret the exact reasons of the research outcome. The author is unable to explain how stimulation of autophagy by HsA facilitate a protective effect while triggering apoptosis.

We thank for your considerate comment and humbly accept it. HsA induced autophagy along with apoptosis. Inhibitior accelerate HsA induced apoptosis. For this reason, we inferred that autophagy has the effect of protecting apoptosis and autophagy inhibition can reduce tumor growth through increase of apoptosis. Previous several reports suggested that inhibition of autophagy enhanced apoptosis [Ref. #4]. We suggest that combination therapy with HsA and autophagy inhibition will be very effective in prostate cancer.

Ref . #4

Sayaka Kanematsu, Norihisa Uehara, Hisanori Miki, Katsuhiko Yoshizawa, Ayako Kawanaka, Takashi Yuri, Airo Tsubura, Autophagy inhibition enhances sulforaphane-induced apoptosis in human breast cancer cells, Anticancer research 30: 3381-3390 (2010)

11. There is a mountain of evidence indicating autophagy induction promotes tumor cell survival under stress conditions and autophagy suppression reduces tumorigenesis (Ref PMID 33946505). However, autophagy is a contest-dependent process that is affected by the stage of tumorigenesis. I have a recommendation to include this point in the discussion.

We agree your thought that Autophagy inhibition can induce tumorigenesis. As mentioned in the previous report, at early stages, autophagy acts as a tumor suppressor mechanism by enhancing the degradation of damaged proteins and organelles, mostly mitochondria and acting as a quality control system that decreases ROS production and genomic instability. On the other hand, at later stages of tumor development, activation of autophagy supplies tumor cells under metabolic stress conditions with nutrients and also maintains mitochondrial metabolism by providing metabolic intermediates, which promote cell survival and tumor growth [Ref #5]. Therefore, we suggest that combination therapy with HsA and autophagy inhibition will be very effective in early therapeutic strategy of prostate cancer. We included that in the discussion section.

Ref

Ref. #5. Yenniffer Ávalos et al., Tumor Suppression and Promotion by Autophagy, BioMed Research International Volume 2014, Article ID 603980, 15 pages.

Round 2

Reviewer 1 Report

The manuscript has been sufficiently improved to warrant publication in Biomolecules.

Author Response

Thank you for your attention.

Reviewer 2 Report

The authors has addressed most of my comments/suggestions.

Author Response

Thank you for your attention.

Reviewer 3 Report

The manuscript is improved and all results are presented will with respective negative and positive control. I will recommend accepting the manuscript.

Thanks 

Author Response

Thank you for your attention.